# Preoperative High C-Reactive Protein to Albumin Ratio Predicts Short- and Long-Term Postoperative Outcomes in Elderly Gastric Cancer Patients

**DOI:** 10.3390/cancers16030616

**Published:** 2024-01-31

**Authors:** Yuki Takemoto, Kazuaki Tanabe, Emi Chikuie, Yoshihiro Saeki, Hiroshi Ota, Nozomi Karakuchi, Akihiro Kohata, Hideki Ohdan

**Affiliations:** 1Department of Gastroenterological and Transplant Surgery, Graduate School of Biomedical and Health Sciences, Hiroshima University, Hiroshima 734-8551, Japan; d203745@hiroshima-u.ac.jp (Y.T.);; 2Department of Perioperative and Critical Care Management, Graduate School of Biomedical and Health Sciences, Hiroshima University, Hiroshima 734-8551, Japan

**Keywords:** complication, C-reactive protein to albumin ratio, elderly patients, gastric cancer, inflammation, nutrition, prognostic factor

## Abstract

**Simple Summary:**

Individual differences exist in the tolerability of surgical intervention among elderly patients, and it is necessary for appropriate treatment planning. The aim of our retrospective study was to investigate the efficacy of markers of preoperative inflammatory and nutritional status and identified perioperative high-risk elderly patients with gastric cancer. We analyzed whether preoperative inflammatory nutritional markers of 571 gastric cancer patients (<65 years old; n = 192, ≥65 years old; n = 379) can predict postoperative short-term and long-term outcomes. Preoperative high C-reactive albumin ratio (CAR) was not only associated with the occurrence of postoperative complications but also with poor prognosis in elderly gastric cancer patients. However, these trends were not observed among younger patients. The importance of these findings is that CAR has the potential to be used as a simple tool to predict short- and long-term postoperative outcomes that reflect the nutritional inflammatory status of the elde.

**Abstract:**

Individualized preoperative assessment of the general condition of elderly patients with gastric cancer is necessary for appropriate surgical treatment planning. This study investigated the efficacy of preoperative markers that could be easily calculated from preoperative peripheral blood to predict the short- and long-term postoperative outcomes of gastrectomy. In total, 571 patients who underwent R0 surgical resection for gastric cancer were enrolled. In the elderly patient group (≥65 years old), univariate analyses revealed that the incidence of postoperative complications was associated with poor performance status (*p* = 0.012), more comorbidities (*p* = 0.020), high C-reactive protein to albumin ratio (CAR, *p* = 0.003), total gastrectomy (*p* = 0.003), open approach (*p* = 0.034), blood transfusion (*p* = 0.002), and advanced cancer (*p* = 0.003). Multivariate analysis showed that a high CAR was associated with a high incidence of postoperative complications (*p* = 0.046). High CAR was also associated with poor OS (*p* = 0.015) and RFS (*p* = 0.035). However, these trends were not observed among younger patients (<65 years old). Preoperative CAR may play a significant role in predicting short- and long-term surgical outcomes, particularly in elderly patients with gastric cancer.

## 1. Introduction

Recently, the average age of patients with gastric cancer has been increasing [1]. Advances in minimally invasive surgical treatments, such as laparoscopic surgery and perioperative management, contribute to good outcomes in elderly patients [2,3], and so the indications for surgical treatment in elderly patients are expanding. However, some reports have shown that elderly patients have more severe complications after gastrectomy compared to younger individuals [4,5,6,7]. Elderly populations generally present a higher prevalence of comorbidities, organ dysfunction, and poorer nutritional status compared to younger individuals. Individual differences exist in the tolerability of surgical intervention among elderly patients, and so individualized preoperative assessment of their general condition is necessary for appropriate treatment planning.

Perioperative inflammatory and nutritional statuses of the host are factors that have been associated with surgical outcomes after some malignancies. Research has been conducted on the effects of inflammation and nutritional status and markers, such as neutrophil to lymphocyte ratio (NLR) [8,9,10], lymphocyte to monocyte ratio (LMR) [11,12,13], C-reactive protein to albumin ratio (CAR) [14,15], and the controlling nutritional status score (CONUT score) [16,17] in patients with gastric cancer, but most of the reports are on patients of all ages, and there are not many reports that focus on the elderly. It is still unclear which markers are effective in elderly gastric cancer patients.

CAR is calculated by dividing serum CRP levels by serum albumin levels. Some authors have suggested that CAR, including protein parameters, is a better indicator of immune response and nutritional status than CRP and albumin alone or neutrophil and lymphocyte assessment of inflammation alone in several diseases, including cancer [18,19,20,21] and infection [22,23]. Several studies have reported that postoperative CAR predicts the occurrence of postoperative complications after gastrectomy [24,25]. However, the value of preoperative CAR remains controversial. In addition, it is unclear whether this index is predictive in the elderly.

In this study, we investigated the efficacy of markers of preoperative inflammatory and nutritional status and identified perioperative high-risk elderly patients with gastric cancer.

## 2. Materials and Methods

### 2.1. Patients

A retrospective cohort study spanning 7 years, from January 2010 to January 2017, was performed, and 709 patients who underwent R0 surgical resection for gastric cancer at the Department of Gastroenterological and Transplant Surgery, Hiroshima University Hospital, were identified. The exclusion criteria were as follows: patients with R1-2 resection cases, emergency cases, remnant gastric cancer cases, cases that did not undergo lymph node dissection (partial resection, etc.), cases with histological types involving special types defined by the Japanese Classification of Gastric Carcinoma, 15th edition [26] (such as carcinoma with lymphoid stroma, adenocarcinoma with enteroblastic differentiation, hepatoid adenocarcinoma, adenocarcinoma of fundic gland type, adenosquamous carcinoma, squamous cell carcinoma, undifferentiated adenocarcinoma), carcinoid tumor and endocrine cell carcinoma, and neuroendocrine carcinoma. Included patients were classified into two groups: an elderly patient group, which included patients aged 65 years or older, and a young patient group, which included patients younger than 65 years of age. 120 cases were excluded according to the exclusion criteria, and 18 were excluded due to missing data. A total of 571 participants were finally included in the analysis. The study was approved by the local institutional review board, and written informed consent was obtained from all patients before treatment.

### 2.2. Surgical Procedure and Follow-Up

We performed gastrectomy according to the Japanese Gastric Cancer Association guidelines for the treatment of gastric cancer [27]. Complications within 30 days of gastrectomy were defined as those of grade II or greater according to the Clavien–Dindo (CD) classification [28]. After gastrectomy, all patients underwent postoperative follow-up to monitor for recurrence. This involved clinical physical examinations, including computed tomography scans and upper gastrointestinal endoscopy, blood chemistry tests, and measurements of tumor markers, specifically carcinoembryonic antigen (CEA) and carbohydrate antigen 19-9 (CA19-9), every 3–6 months for at least 5 years or until death. 

### 2.3. Preoperative Assessments

The measured variables included age, sex, American Society of Anesthesiologists physical status (ASA-PS), body mass index (BMI), and comorbidities. Comorbidities were described using the Charlson Comorbidity Index (CCI) [29]. Laboratory data were collected preoperatively and calculated to assess preoperative inflammation and nutritional status. These cut-off values were determined using receiver operating characteristic (ROC) curve analysis. CONUT score was calculated based on a published paper [30].

### 2.4. Preoperative Predictive Scoring Model

To improve the accuracy of prediction, we propose a predictive CAR-based model, combined with preoperative ASA-PS and surgical procedures, which exert a substantial influence on short- and long-term outcomes following gastrectomy. Each preoperative factor was assigned a score of one: CAR ≥ 0.024, ASA-PS 3, and total gastrectomy. These indices had similar odds ratios in the previous analysis of preoperative complications (1.62, 1.49, and 1.62, respectively) and gave equal points. The total scores were divided into two groups: score 0–1 (low-risk group) and 2–3 (high-risk group).

### 2.5. Statistical Analysis

Data were analyzed using the JMP Pro statistical software package 15.0.0 (SAS Institute, Cary, NC, USA). Continuous variables were expressed as medians and ranges. Nominal variables were expressed as numbers (%). Nonparametric quantitative data and nominal variables were analyzed by the Mann–Whitney U test and the chi-squared test, respectively. We used multivariate logistic regression analysis to identify risk factors for postoperative complications.

Overall survival (OS) and recurrence-free survival (RFS) rates were calculated using the Kaplan–Meier method and the log-rank test. Multivariate analyses of the OS and RFS were performed using Cox regression models. 

Propensity score matching analysis was performed using logistic regression analysis to create a propensity score for the low CAR group and the high CAR group with a logistic regression model. The following variables were entered into the propensity score model: age, sex, comorbidities (CCIs), operative procedures, and pathological stage. One-to-one matching without replacement was performed with a caliper width of 0.25, and the resulting score-matched pairs were used in subsequent analyses.

Statistical significance was set at *p* < 0.05.

## 3. Results

### 3.1. Comparison of Clinicopathological Characteristics between Young and Elderly Patient Groups

During the study period, 379 (66.4%) and 192 (33.6%) patients were classified into the elderly (≥65 years of age) and young (<65 years of age) patient groups, respectively. The clinicopathological characteristics and surgical and pathological outcomes were compared between the groups (Table 1). Elderly patients had poorer ASA-PS (*p* < 0.001), more comorbidities (*p* < 0.001), lower preoperative albumin (*p* < 0.001), and total cholesterol (*p* = 0.003) levels, and higher CEA (*p* < 0.001) and CRP (*p* < 0.001) levels compared to young patients. The preoperative LMR (*p* = 0.004), CAR (*p* < 0.001), and CONUT scores (*p* = 0.001) of elderly patients indicated high inflammation and poor nutritional status. There were no significant differences in surgical technique, approach, operative time, or intraoperative blood transfusion between the two groups; however, the incidence of postoperative complications (≥CD II) was higher (*p* = 0.006), and the duration of hospital stay was longer (*p* < 0.001) in the elderly group. Pathologically, a more differentiated type was observed in the elderly patient group, with no differences in the location and depth of the tumor, frequency of metastasis to regional lymph nodes, and pathologic progression.

### 3.2. Identification and Comparison of Risk Factors for Postoperative Complications between Young and Elderly Patient Groups

A comparison between the elderly and young patient groups showed that elderly patients with gastric cancer were preoperatively undernourished and had a higher incidence of postoperative complications. Univariate analysis of risk factors associated with ≥CD II postoperative complications, involving all cases, revealed that age (*p* = 0.001), ASA-PS (*p* = 0.005), CCI (*p* < 0.001), preoperative lymphocyte (*p* = 0.031), albumin (*p* < 0.001), CRP (*p* = 0.020), total cholesterol (*p* = 0.003), CEA (*p* = 0.005), LMR (*p* = 0.005), CAR (*p* = 0.010), CONUT scores (*p* = 0.012), surgical procedure (*p* < 0.001), surgical approach (*p* = 0.005), intraoperative bleeding (*p* < 0.001), blood transfusion (*p* < 0.001), depth of the tumor (*p* = 0.010), and pathological stage (*p* = 0.040) were significantly associated with the incidence of complications (Table 2). 

Based on the risk factors identified in the all-cases analysis, we performed univariate and multivariate analyses of the elderly and young patient groups, respectively. In the elderly patient group, univariate analyses revealed that the incidence of postoperative complications was significantly associated with ASA-PS 3 (*p* = 0.012), ≥1 comorbidities (*p* = 0.020), high CAR (*p* = 0.003), total gastrectomy (*p* = 0.003), open approach (*p* = 0.034), intraoperative blood transfusion (*p* = 0.002), and pathological stage II and III (*p* = 0.003). Multivariate analyses revealed that high preoperative CAR (Odds ratio [OR] 1.62; 95% confidence interval [CI] 1.01–2.62; *p* = 0.046) was independently related to the incidence of postoperative complications (Table 3a). In the young patient group, two preoperative inflammatory nutritional markers, LMR and CONUT score (*p* = 0.025 and 0.022, respectively), were identified as risk factors for complications in the univariate analysis; however, multivariate analysis revealed no association between preoperative inflammatory nutritional markers and postoperative complications (Table 3b).

### 3.3. Prognostic Significance of Factors Identified as Risk Factors for Postoperative Complications

Patients with postoperative complications had worse 5-year OS and RFS in both the elderly (OS, *p* = 0.019; RFS, *p* = 0.020) and young patient groups (OS, *p* = 0.013; RFS, *p* = 0.016) (Figure 1). 

Univariate and multivariate analyses were performed to investigate the possibility that risk factors for postoperative complications also had a negative effect on OS and RFS in the elderly and young patient groups, respectively. Table 4a shows the association between each risk factor and OS according to age group. In the elderly patient group, ASA-PS 3 (*p* < 0.001), ≥1 comorbidities (*p* < 0.001), high CEA (*p* = 0.012), low LMR (*p* < 0.001), high CAR (*p* < 0.001), high CONUT score (*p* = 0.004), total gastrectomy (*p* < 0.001), open approach (*p* < 0.001), intraoperative blood transfusion (*p* = 0.003), and pathological stage II or III (*p* < 0.001) were significantly associated with poor OS in the univariate analysis. In the multivariate analysis, high CAR (hazard ratio [HR] 2.02; 95% CI 1.51–3.56; *p* = 0.015) was identified as an independent risk factor for poor OS with ASA-PS 3 (HR 2.34; 95% CI 1.21–4.55; *p* = 0.012), ≥1 comorbidities (HR 2.90; 95% CI 1.45–5.79; *p* = 0.003), total gastrectomy (HR 1.92; 95% CI 1.06–3.50; *p* = 0.033), and open approach (HR 1.92; 95% CI 1.00–3.69; *p* = 0.049). In contrast, in the young patient group, inflammatory nutritional markers were not associated with OS; only surgical and pathological factors, including: total gastrectomy (HR 4.28; 95% CI 1.20–15.24; *p* = 0.025), intraoperative blood transfusion (HR 6.41; 95% CI 1.31–31.44; *p* = 0.022), and pathological stage II or III (HR 12.58; 95% CI 2.19–72.18; *p* = 0.005). Analysis of prognostic factors for RFS (Table 4b) showed that high CAR (HR 2.51; 95% CI 1.34–4.67; *p* = 0.035), ASA-PS 3 (HR 2.51; 95% CI 1.34–4.67; *p* = 0.004), ≥1 comorbidities (HR 2.42; 95% CI 1.29–4.52; *p* = 0.006), and open approach (HR 2.51; 95% CI 1.34–4.67; *p* = 0.035) were also significantly associated with RFS in the elderly, but inflammatory nutritional markers were not prognostic factors in the young group, neither in OS nor in RFS.

### 3.4. Efficacy of CAR in Predicting the Incidence of Postoperative Complications and Long-Term Prognosis in the Elderly Patient Group after Propensity Score Matching

Preoperative high CAR predicted the postoperative complications (*p* = 0.004; Table 5) and poor OS and RFS (*p* < 0.001 and *p* < 0.001, respectively; Figure 2a) among elderly patients with gastric cancer. In the elderly patient group, patients with a high CAR had significantly poorer ASA-PS scores (*p* = 0.002), more comorbidities (*p* = 0.006), and more advanced pathological stages (*p* = 0.012). Propensity score matching analysis also showed that a high CAR was a risk factor for postoperative complications (*p* = 0.039, Table 5) and was associated with significantly worse OS (*p* = 0.003) and RFS (*p* = 0.011) compared to those with a low CAR (Figure 2b).

### 3.5. Preoperative Predictive Scoring Model for Elderly Gastric Cancer Patients Based on CAR, ASA-PS, Surgical Procedures

Preoperative CAR was a significant predictive factor for the incidence of postoperative complications, as well as OS and RFS. To improve the accuracy of prediction, we proposed a predictive CAR-based model, combined with preoperative ASA-PS and surgical procedures, which exert a substantial influence on short- and long-term outcomes following gastrectomy.

According to the prediction score model we presented, a total of 379 elderly gastric cancer patients were classified into two groups: 68 in the high-risk group and 311 in the low-risk group. In comparison to the low-risk group, the high-risk group had an increased incidence of postoperative complications (48.5% vs. 23.8%, *p* < 0.001), and the same was true after propensity score matching analysis considering sex, comorbidities, and pathological stages (48.5% vs. 26.5%, *p* = 0.008, Figure 3). Furthermore, the prediction score model was useful in predicting the long-term prognosis of elderly patients with gastric cancer (Figure 4).

## 4. Discussion

In this study, we investigated the effects of preoperative CAR on postoperative complications and long-term prognosis after curative gastrectomy in patients under 65 years of age and 65 years of age or older. In the group ≥65 years, high preoperative CAR was identified as a risk factor for the incidence of ≥CD II postoperative complications and poor OS and RFS. This association was not found in the younger patient group. 

To the best of our knowledge, this is the first study to report that a high preoperative CAR is strongly associated with short-term postoperative outcomes and long-term prognosis in elderly patients with gastric cancer.

CAR was related to surgical outcomes in the older patient group only. Thus, CAR may be an effective screening tool in older groups that also include more frail patients. Conversely, the younger group displayed a relatively homogeneous nutritional status, suggesting that CAR might have a diminished impact on surgical outcomes in this population. Preoperative CAR levels were strongly associated with age and poor ASA-PS (Appendix A), leading us to believe that it is more effective in evaluating elderly people with diverse immune and nutritional conditions.

Age-related increases in inflammatory molecules, such as interleukin-6 (IL-6) levels, are associated with several pathophysiological processes, including atherosclerosis, osteoporosis, and sarcopenia, as well as functional decline, disability, and all-cause mortality in older adults [31].

Chronic inflammation and malnutrition of the host may influence both the acute phase immune response to perioperative insults and the long-term postoperative antitumor response. We postulated the following mechanism as the reason why chronic inflammation and malnutrition, as reflected by high CAR levels, influence the occurrence of early postoperative complications: during the acute invasion phase, such as surgery, an intensified preoperative activation of pro-inflammatory cascades can increase the susceptibility of the host to infections and immune overreactions, resembling a systematic inflammatory response syndrome [32]. It is known that an imbalance in the immune response caused by chronic inflammation due to aging, cancer, and comorbidities, such as obesity and diabetes, leads to systemic inflammatory dysregulation in the perioperative period, resulting in fatal complications. Immune dysregulation is complexly related to dysregulation in each phase of the host immune response to surgical invasion, and IL6 is known to play an important role in immune activation and suppression [33]. The previous report suggested that upregulation of the TLR/NFκB/IL-6 pathway in monocytes was observed in the development of SARS after abdominal surgery [34]. CRP produced in the liver and adipocytes by IL-6 may be useful in measuring and predicting systemic inflammatory dysregulation [33]. In addition, perioperative malnutrition leads to decreased immune competence, delayed wound healing, and susceptibility to infection [35,36,37]. 

On the other hand, long-term chronic inflammation, caused by comorbidities and aging, can predispose the host to various chronic diseases, including cancer [38]. The relationship between high CAR levels and poor prognosis has been reported in several malignancies [18,38,39,40]. IL-6 is involved in the proliferation and differentiation of various malignant tumor cells through the IL-6-Jak-Stat signaling pathway [31,41]. Tumor-induced cytokines such as TNF-α, interleukin 1 beta, and IL-6 contribute to the inhibition of albumin gene expression in cachexia [42]. Although there are few studies that directly explain whether age-related chronic inflammation directly affects gastric cancer progression, there are many studies on the biological dynamics that chronic inflammation caused by Helicobacter pylori infection has on carcinogenesis and progression. Piazuelo MB et al. summarize the mechanism by which chronic inflammation affects gastric cancer progression [43]. They claimed that H. pylori reacts with gastric epithelial cells and produces many proinflammatory cytokines and chemokines, including IL-1β, IL-6, IL-8, and IL-18. These cytokines contribute to the development of gastric cancer by acquiring anti-apoptotic properties, causing DNA damage and genetic instability through activation of the JAK/STAT and NFkB pathways. Various cytokines induce changes in tumor cells and surrounding immune cells, fibroblasts, and endothelial cells, forming the tumor microenvironment. Unresolved chronic inflammation caused by Helicobacter pylori promotes the secretion of cytokines and chemokines from tumor cells and recruit cells such as MDSCs, Tregs, and TAMs that are beneficial for tumor survival [44]. These cells suppress the function of NK cells and CD8+ T cells and promote tumor growth and invasion. These mechanisms of inflammation and tumor growth caused by Helicobacter pylori are not directly applicable to systemic chronic inflammation caused by senile frailty. However, some similar pathways may be at play. 

Malnutrition is also associated with impaired immune function against cancer progression, poor response to therapy, and increased susceptibility to treatment-related adverse events [45]. Host’s chronic inflammation and malnutrition was reflected in a high CAR, which may predict tumor recurrence, decreased tolerability of treatment, and increased risk of other lethal complications that negatively impact long-term prognosis.

Thus, as a combination of CRP and albumin, CAR may better reflect the severity of inflammation in frail older adults.

We considered other commonly used indicators of general health in the patients included in this study. It was confirmed that poor ASA-PS was a risk factor for short- and long-term outcomes. ASA-PS was an important element along with CAR, and we placed importance on it because it could be expected to improve with preoperative intervention. We hoped that by combining the clinical index ASA-PS and the measured value of CAR, we would be able to identify high-risk patients more accurately. In our analysis population, preoperative BMI was not found to be statistically associated with outcome. This might be due to the relatively low proportion of advanced stages leading to eating disorders and weight loss in the target population (pStage II/III; n = 142, 24.9% vs. pStage I; n = 429, 75.1%). Furthermore, preoperative weight loss is an interesting factor but unfortunately, it was often not recorded and could not be included in this study. There have been reports that the Geriatric Nutritional Risk Index (GNRI), an assessment index using body weight, was associated with prognosis in elderly gastric cancer patients [46], but our analysis found no association (data not published in the paper). This might also be attributed to the large proportion of patients with early gastric cancer who did not have significant weight loss. No statistical relationship was found for preoperative hemoglobin levels with outcomes, so we investigated whether intraoperative blood transfusion was performed. However, intraoperative blood transfusion might be associated with various confounding factors. Univariate analysis showed a statistical relationship between transfusion and surgical outcomes, but multivariate analysis did not in the elderly patient group.

Our predictive model using preoperative CAR levels in elderly patients may help identify at-risk individuals and plan appropriate treatments. Our predictive model included three risk factors: preoperative high-CAR, total gastrectomy, and poor ASA-PS (score 3). Patients with more than two risk factors had a high risk of postoperative complications and poor prognosis. Improvements in risk scores may contribute to better outcomes in high-risk patients. Adequate preoperative anti-inflammatory and nutritional treatments may be able to decrease preoperative CAR levels. Several reports, using anti-inflammatory drugs in the perioperative period to control systemic inflammation, have suggested that the intraoperative administration of nonsteroidal anti-inflammatory drugs improved prognosis in patients with breast, kidney, and lung cancer [47]. Additionally, the administration of long-term low-dose aspirin has been associated with improved prevention of cancer [48,49]. Forget P et al. reported that NSAIDs’ beneficial effect was greater in breast cancer patients with a higher NLR [47]. Other candidate anti-inflammatory strategies, including cytokine antagonists such as IL-6 and TNF [43,50], have been reported. However, none of these have been clinically applied to cancer, and future verification is expected. We did not use anti-inflammatory drugs for systematic inflammation control in the preoperative period. Although it is unclear whether these anti-inflammatory drugs contribute to the reduction of CAR, it is an interesting topic for future research. Preoperative treatment of comorbidities can improve the ASA-PS. Total gastrectomy is generally associated with a high risk of postoperative complications and is unfavorable for postoperative nutritional status. In the elderly, the nutritional disadvantages of total gastrectomy outweigh its oncological benefits in some cases. These results may be useful for selecting preoperative systemic care and surgical treatment plans.

This study has some limitations. First, this was a retrospective study conducted at a single institution with a relatively small sample size. We think that it is necessary to study this in a multi-institutional setting or with other groups. Second, definite cutoff values for CAR are yet to be defined. Our adopted cutoff value was 0.024, which was calculated using ROC curve analysis. However, the area under the ROC curve was only 0.57, indicating that the accuracy of the cutoff was relatively poor. In other studies, the CAR cutoff value was set at 0.028 in a lung cancer study [20], 0.037 in a hepatocellular carcinoma study [40], and 0.038 in a colorectal cancer study [51]. Although our cutoff value was close to these values, additional studies are still warranted.

## 5. Conclusions

Preoperative CAR can play a significant role in predicting short- and long-term surgical outcomes among elderly patients with gastric cancer. Furthermore, CAR emerges as an independent and crucial tool, aiding in the development of optimal therapeutic strategies for operable elderly patients with gastric cancer.

## Figures and Tables

**Figure 1 cancers-16-00616-f001:**
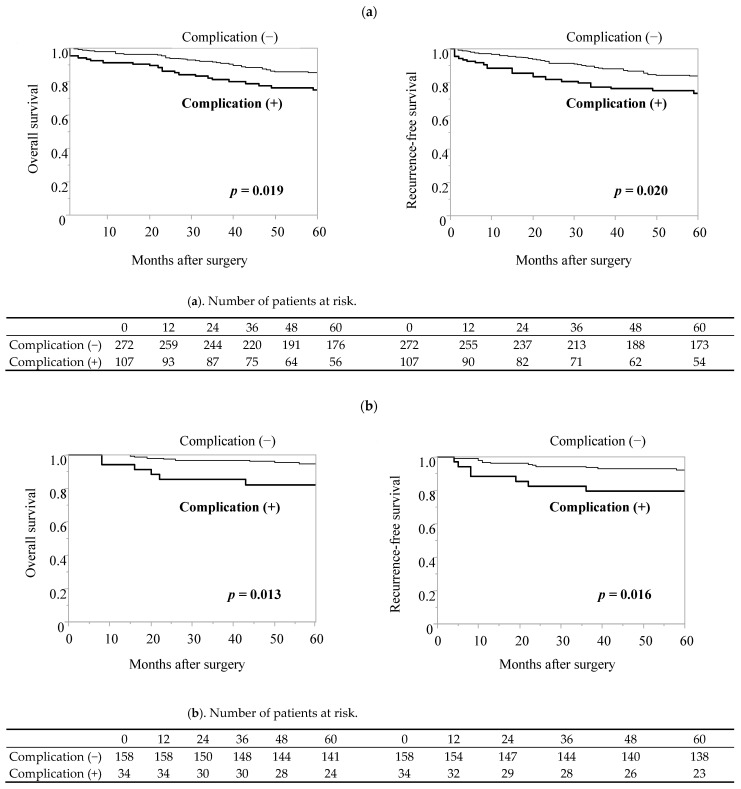
Kaplan–Meier analysis of survival of patients with gastric cancer stratified based on being with or without ≥Clavien–Dindo classification II postoperative complications. (**a**) Elderly patient group, (**b**) young patient group.

**Figure 2 cancers-16-00616-f002:**
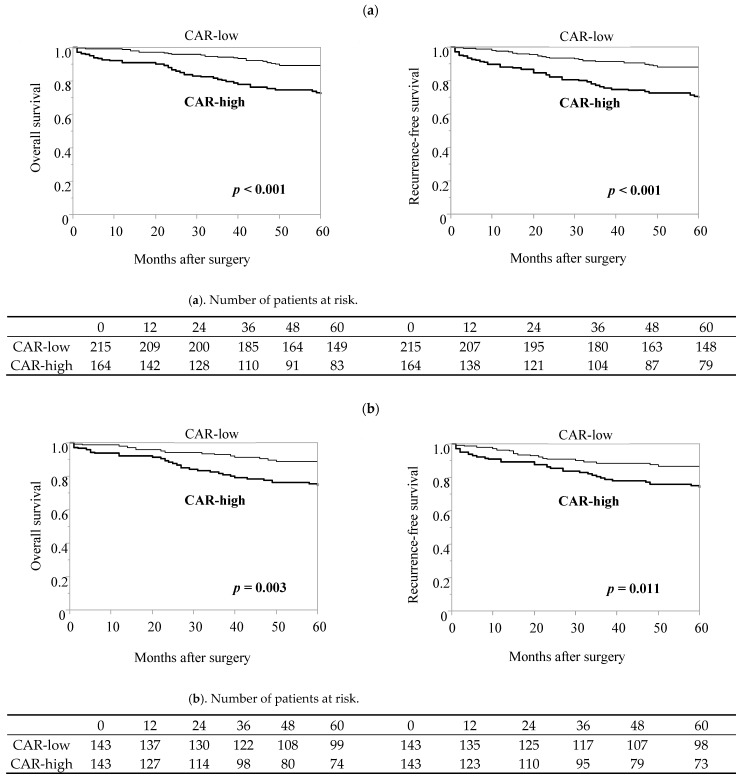
Kaplan–Meier analysis of survival in elderly patients with gastric cancer, stratified by high or low CAR. (**a**) Entire cohort of elderly patients, (**b**) propensity-matched pairs. Propensity score-matching analysis was performed using logistic regression analysis to create a propensity score for the low-score group and high-score group with a logistic regression model. The following variables were entered into the propensity score model: sex, ASA-PS, CCI, procedure, approach, and pStage. One-to-one matching without replacement was performed with a 0.25 caliper width, and the resulting score-matched pairs were used in subsequent analyses (n = 143).

**Figure 3 cancers-16-00616-f003:**
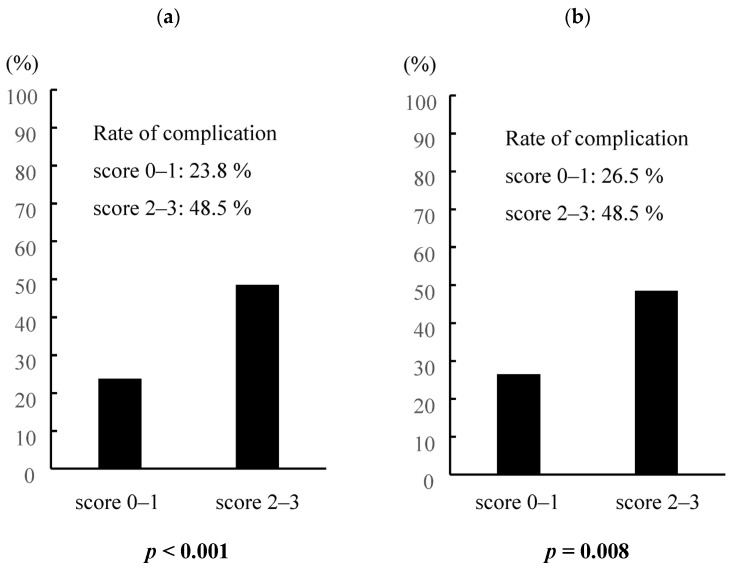
Predictive score system for postoperative complications among elderly gastric cancer patients. (**a**) Entire cohort of elderly patients, (**b**) propensity-matched pairs. Propensity score-matching analysis was performed using logistic regression analysis to create a propensity score for the low-score group and the high-score group with a logistic regression model. The following variables were entered into the propensity score model: sex, CCI, approach, and pStage. One-to-one matching without replacement was performed with a 0.25 caliper width, and the resulting score-matched pairs were used in subsequent analyses (n = 68).

**Figure 4 cancers-16-00616-f004:**
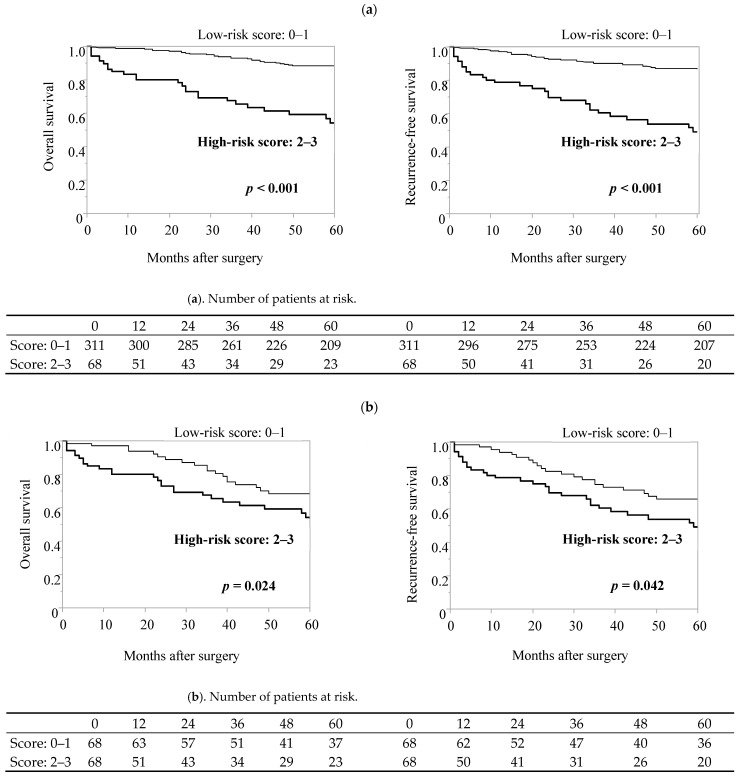
The predicting score system for postoperative complications among elderly gastric cancer patients was useful in predicting long-term prognosis. (**a**) Entire cohort of elderly patients, (**b**) propensity-matched pairs. Propensity score-matching analysis was performed using logistic regression analysis to create a propensity score for the low-score group and the high-score group with a logistic regression model. The following variables were entered into the propensity score model: sex, CCI, approach, and pStage. One-to-one matching without replacement was performed with a 0.25 caliper width, and the resulting score-matched pairs were used in subsequent analyses (n = 68).

**Table 1 cancers-16-00616-t001:** Comparison of clinicopathological characteristics between elderly and young patient groups.

	All Patients (N = 571)	Elderly (≥65) Patient Group (n = 379)	Young (<65) Patient Group (n = 192)	*p*-Value
Sex				0.240
Male	393 (68.8)	267 (70.5)	126 (65.6)	
Female	178 (31.2)	112 (29.5)	66 (34.4)	
BMI	22.1 (14.5–32.9)	22.3 (14.5–31.5)	22.0 (15.1–32.9)	0.670
ASA-PS				**<0.001**
1	61 (10.7)	32 (8.5)	29 (15.1)	
2	470 (82.3)	309 (81.5)	161 (83.9)	
3	40 (7.0)	38 (10.0)	2 (1.0)	
Charlson Comorbidity Index *				**<0.001**
Low: 0	326 (57.1)	167 (44.1)	159 (82.8)	
Medium: 1–2	187 (32.8)	158 (41.7)	29 (15.1)	
High: 3–4	47 (8.2)	44 (11.6)	3 (1.6)	
Very high: ≥5	11 (1.9)	10 (2.6)	1 (0.5)	
Preoperative evaluation				
WBC (μL)	5600 (2300–15,200)	5590 (2300–15,200)	5650 (2420–11,960)	0.512
Ne (μL)	3339 (930–12,070)	3317 (930–12,070)	3401 (1111–11,362)	0.277
Ly (μL)	1670 (120–5426)	1679 (280–5426)	1651 (120–3902)	0.987
Mo (μL)	319 (69–1376)	333 (99–984)	288 (69–1376)	**<0.001**
Alb (g/dL)	4.1 (2.2–5.3)	3.6 (2.2–5.1)	3.9 (2.3–5.3)	**<0.001**
CRP (mg/L)	0.07 (0.02–11.90)	0.07 (0.02–11.90)	0.05 (0.02–3.93)	**<0.001**
Total cholesterol (mg/mL)	191 (92–321)	189 (92–320)	198 (116–321)	**0.003**
CEA (ng/mL)	2.2 (0.5–64.0)	2.3 (0.5–64.0)	1.8 (0.5–14.7)	**<0.001**
CA19-9 (U/mL)	5.0 (0.04–7775.0)	5.0 (0.04–7775.0)	5.0 (2.0–182.0)	0.213
Preoperative nutrition and inflammation markers				
NLR	1.99 (0.51–95.00)	1.97 (0.51–13.80)	2.01 (0.65–95.00)	0.435
LMR	5.15 (0.25–22.07)	4.97 (0.86–17.19)	5.80 (0.25–22.08)	**0.004**
CAR	0.016 (0.004–3.838)	0.019 (0.004–3.838)	0.012 (0.004–0.914)	**<0.001**
CONUT score				**0.001**
Normal (0–1)	351 (61.5)	213 (56.2)	138 (71.9)	
Light malnutrition (2–4)	182 (31.9)	134 (35.4)	48 (25.0)	
Moderate malnutrition (5–8)	34 (5.9)	30 (7.9)	4 (2.1)	
Severe malnutrition (9–12)	4 (0.7)	2 (0.5)	2 (1.0)	
Procedure				0.569
DG	378 (66.2)	247 (65.2)	131 (68.2)	
TG	131 (22.9)	92 (24.3)	39 (20.3)	
PG	54 (9.5)	36 (9.5)	18 (9.4)	
PPG	8 (1.4)	4 (1.0)	4 (2.1)	
Approach				0.101
Laparoscopy	357 (62.5)	228 (60.2)	129 (67.2)	
Open	214 (37.5)	151 (39.8)	63 (32.8)	
Operative time (min)	298 (142–885)	301 (142–749)	293 (147–885)	0.598
Intraoperative bleeding (mL)	56 (0–6882)	60 (0–6882)	50 (4–1550)	0.280
Intraoperative blood transfusion				0.130
No	542 (94.9)	356 (93.9)	186 (96.9)	
Yes	29 (5.1)	23 (6.1)	6 (3.1)	
Postoperative complications (≥CD II)				**0.006**
Absent	430 (75.3)	272 (71.8)	158 (82.3)	
Present	141 (24.7)	107 (28.2)	34 (17.7)	
Hospital stays (days)	12 (7–344)	13 (8–136)	11 (7–344)	**<0.001**
Location of tumor				0.065
Upper	143 (25.0)	102 (26.9)	41 (21.4)	
Middle	207 (36.3)	125 (33.0)	82 (42.7)	
Low	221 (38.7)	152 (40.1)	69 (35.9)	
Histopathological type **				**<0.001**
Differentiated	281 (49.2)	218 (57.5)	63 (32.8)	
Undifferentiated	290 (50.8)	161 (42.5)	129 (67.2)	
Depth of tumor ***				0.550
T1a,b	393 (68.8)	254 (67.0)	139 (72.4)	
T2	65 (11.4)	45 (11.9)	20 (10.4)	
T3	60 (10.5)	41 (10.8)	19 (9.9)	
T4a,b	53 (9.3)	39 (10.3)	14 (7.3)	
Lymph node metastasis ***				0.281
N0	430 (75.3)	279 (73.6)	151 (78.6)	
N1	61 (10.7)	44 (11.6)	17 (8.9)	
N2	40 (7.0)	31 (8.2)	9 (4.7)	
N3	40 (7.0)	25 (6.6)	15 (7.8)	
Pathological stage ***				0.536
I	429 (75.1)	280 (73.9)	149 (77.6)	
II	61 (10.7)	41 (10.8)	20 (10.4)	
III	81 (14.2)	58 (15.3)	23 (12.0)	

Variables in bold are statistically significant (*p* < 0.05). Categorical variables were described using frequencies and percentages, and continuous variables were described using median and range. * Charlson Comorbidity Index was defined by only comorbidities without adjusting age. ** Differentiated type and undifferentiated type involved papillary carcinoma, tubular adenocarcinoma, and poorly differentiated adenocarcinoma, mucinous adenocarcinoma, signet-ring cell carcinoma, respectively. *** According to the 15th edition of the Japanese Classification of Gastric Carcinoma. Abbreviations: Alb, albumin; ASA-PS, American society of anesthesiologists physical status; BMI, body mass index; CA19-9, carbohydrate antigen 19-9; CAR, C-reactive protein/albumin ratio; CD, Clavien–Dindo classification; CEA, carcinoembryonic antigen; CONUT score, controlling nutritional status score; CRP, C-reactive protein; DG, distal gastrectomy; LMR, lymphocyte-to-monocyte ratio; Ly, lymphocytes; Mo, monocyte; Ne, neutrophil; NLR, neutrophil-to-lymphocyte ratio; PG, proximal gastrectomy; PPG, pylorus-preserving gastrectomy; TG, total gastrectomy; WBC, white blood cell.

**Table 2 cancers-16-00616-t002:** Comparison of clinicopathological features between postoperative complications (≥CD II) in absent and present cases.

	All Patients (N = 571)	Complication (−) (n = 430)	Complication (+) (n = 141)	*p*-Value
Age	69 (21–89)	68 (21–89)	71 834–89)	**0.001**
Sex				0.407
Male	393 (68.8)	292 (67.9)	101 (71.6)	
Female	178 (31.2)	138 (32.1)	40 (28.4)	
BMI	22.1 (14.5–32.9)	22.1 (14.5–32.9)	22.1 (15.5–31.5)	0.711
ASA-PS				**0.005**
1	61 (10.7)	50 (11.6)	11 (7.8)	
2	470 (82.3)	358 (83.3)	112 (79.4)	
3	40 (7.0)	22 (5.1)	18 (12.8)	
Charlson Comorbidity Index *				**<0.001**
Low: 0	326 (57.1)	262 (60.9)	64 (45.4)	
Medium: 1–2	187 (32.8)	134 (31.2)	53 (37.6)	
High: 3–4	47 (8.2)	25 (5.8)	22 (15.6)	
Very high: ≥5	11 (1.9)	9 (2.1)	2 (1.4)	
Preoperative evaluation				
WBC (μL)	5600 (2300–15,200)	5600 (2300–13,890)	5620 (2420–15,200)	0.299
Ne (μL)	3339 (930–12,070)	3329 (979–12,070)	3347 (930–9634)	0.749
Ly (μL)	1670 (120–5426)	1711 (120–4681)	1592 (266–5426)	**0.031**
Mo (μL)	319 (69–1376)	315 (69–1376)	332 (110–983)	0.099
Alb (g/dL)	4.1 (2.2–5.3)	4.2 (2.4–5.3)	4.0 (2.2–5.1)	**<0.001**
CRP (mg/L)	0.07 (0.02–11.90)	0.06 (0.02–11.90)	0.10 (0.02–4.04)	**0.020**
Total cholesterol (mg/mL)	191 (92–321)	193 (112–321)	185 (92–286)	**0.003**
CEA (ng/mL)	2.2 (0.5–64.0)	2.0 (0.5–64.0)	2.4 (0.5–23.5)	**0.005**
CA19-9 (U/mL)	5.0 (0.04–7775.0)	5.0 (0.04–7775.0)	6.0 (2.0–429.0)	0.328
Preoperative nutrition and inflammation markers				
NLR	1.99 (0.51–95.00)	1.96 (0.52–95.00)	2.05 (0.51–13.83)	0.196
LMR	5.15 (0.25–22.07)	5.41 (0.25–22.07)	4.72 (0.48–11.26)	**0.005**
CAR	0.016 (0.004–3.838)	0.014 (0.004–3.839)	0.024(0.004–1.154)	**0.010**
CONUT score				**0.012**
Normal (0–1)	351 (61.5)	277 (64.4)	74 (52.5)	
Light malnutrition (2–4)	182 (31.9)	129 (30.0)	53 (37.6)	
Moderate malnutrition (5–8)	34 (5.9)	23 (5.4)	11 (7.8)	
Severe malnutrition (9–12)	4 (0.7)	1 (0.2)	3 (2.1)	
Procedure				**<0.001**
DG	378 (66.2)	299 (69.5)	79 (56.0)	
TG	131 (22.9)	82 (19.1)	49 (34.8)	
PG	54 (9.5)	45 (10.5)	9 (6.4)	
PPG	8 (1.4)	4 (0.9)	4 (2.8)	
Approach				**0.005**
Laparoscopy	357 (62.5)	283 (65.8)	74 (52.5)	
Open	214 (37.5)	147 (34.2)	67 (47.5)	
Operation time (min)	298 (142–885)	298 (147–885)	301 (142–755)	0.209
Intraoperative bleeding (mL)	56 (0–6882)	50 (0–2870)	80 (8–6882)	**<0.001**
Intraoperative blood transfusion				**<0.001**
No	542 (94.9)	418 (97.2)	124 (87.9)	
Yes	29 (5.1)	12 (2.8)	17 (12.1)	
Hospital stays (days)	12 (7–344)	11 (7–35)	22 (9–61)	**<0.001**
Location of tumor				0.532
Upper	143 (25.0)	103 (24.0)	40 (28.4)	
Middle	207 (36.3)	160 (37.2)	47 (33.3)	
Low	221 (38.7)	167 (38.8)	54 (38.3)	
Histopathological type **				0.371
Differentiated	281 (49.2)	207 (48.1)	74 (52.5)	
Undifferentiated	290 (50.8)	223 (51.9)	67 (47.5)	
Depth of tumor ***				**0.010**
T1a,b	393 (68.8)	310 (72.1)	83 (58.9)	
T2	65 (11.4)	47 (10.9)	18 (12.8)	
T3	60 (10.5)	36 (8.4)	24 (17.0)	
T4a,b	53 (9.3)	37 (8.6)	16 (11.4)	
Lymph node metastasis ***				0.232
N0	430 (75.3)	332 (77.2)	98 (69.5)	
N1	61 (10.7)	40 (9.3)	21 (14.9)	
N2	40 (7.0)	29 (6.7)	11 (7.8)	
N3	40 (7.0)	29 (6.7)	11 (7.8)	
Pathological stage ***				**0.040**
I	429 (75.1)	334 (77.7)	95 (67.4)	
II	61 (10.7)	39 (9.1)	22 (15.6)	
III	81 (14.2)	57 (13.3)	24 (17.0)	

Variables in bold are statistically significant (*p* < 0.05). Categorical variables were described using frequencies and percentages, and continuous variables were described using median and range. * Charlson Comorbidity Index was defined by only comorbidities without adjusting age. ** Differentiated type and undifferentiated type involved papillary carcinoma, tubular adenocarcinoma, and poorly differentiated adenocarcinoma, mucinous adenocarcinoma, signet-ring cell carcinoma, respectively. *** According to the 15th edition of the Japanese Classification of Gastric Carcinoma. Abbreviations: Alb, albumin; ASA-PS, American society of anesthesiologists physical status; BMI, body mass index; CA19-9, carbohydrate antigen 19-9; CAR, C-reactive protein/albumin ratio; CD, Clavien–Dindo classification; CEA, carcinoembryonic antigen; CONUT score, controlling nutritional status score; CRP, C-reactive protein; DG, distal gastrectomy; LMR, lymphocyte-to-monocyte ratio; Ly, lymphocytes; Mo, monocyte; Ne, neutrophil; NLR, neutrophil-to-lymphocyte ratio; PG, proximal gastrectomy; PPG, pylorus-preserving gastrectomy; TG, total gastrectomy; WBC, white blood cell.

**Table 3 cancers-16-00616-t003:** Univariate and multivariate analyses to assess the risk factors for postoperative complications in elderly and young gastric cancer patient groups. (**a**) analyses of the risk factors for postoperative complications (≥CD II) in the elderly patient group. (**b**) Analyses of the risk factors for postoperative complications (≥CD II) in the young patient group.

**(a)**										
**Elderly Patient Group: n = 379**			**Univariate Analysis**				**Multivariate Analysis**			
**Variables**	**Categories**	**Number of Patients** **with Complication (%)**	**95% CI of OR**				**95% CI of OR**			
			**OR**	**Low**	**High**	***p*-Value**	**OR**	**Low**	**High**	***p*-Value**
ASA-PS	3	17 (46.0)	2.38	1.19	4.74	**0.012**	1.49	0.71	3.15	0.291
	1, 2	90 (26.3)								
Charlson Comorbidity Index *	≥Medium risk	70 (33.0)	1.73	1.09	2.75	**0.020**	1.48	0.90	2.43	0.118
	Low	37 (22.2)								
CEA (ng/mL)	≥5.0	18 (36.7)	1.52	0.81	2.85	0.195				
	<5.0	87 (27.7)								
LMR ***	<5.08	63 (32.0)	1.47	0.94	2.32	0.092				
	≥5.08	44 (24.2)								
CAR ***	≥0.024	59 (36.0)	1.95	1.24	3.07	**0.003**	1.62	1.01	2.62	**0.046**
	<0.024	48 (22.3)								
CONUT score	≥Light malnutrition	52 (31.3)	1.31	0.84	2.05	0.238				
	Normal	55 (25.8)								
Procedure	TG	37 (40.2)	2.09	1.27	3.43	**0.003**	1.62	0.92	2.84	0.096
	DG, PG, PPG	70 (24.4)								
Approach	Open	51 (33.8)	1.56	1.01	2.46	**0.034**	0.80	0.45	1.43	0.455
	Laparoscopy	56 (24.6)								
Intraoperative blood transfusion	Yes	13 (56.5)	3.62	1.54	8.54	**0.002**	2.11	0.83	5.42	0.118
	No	94 (26.4)								
Pathological stage **	II, III	39 (39.8)	2.07	1.27	3.37	**0.003**	1.70	0.95	3.05	0.076
	I	68 (24.2)								
**(b)**										
**Young Patient Group: n = 192**			**Univariate Analysis**				**Multivariate Analysis**			
**Variables**	**Categories**	**Number of Patients** **with Complication (%)**	**95% CI of OR**				**95% CI of OR**			
			**OR**	**Low**	**High**	***p*-Value**	**OR**	**Low**	**High**	***p*-Value**
ASA-PS	3	1 (50.0)	4.75	0.29	0.32	0.229				
	1, 2	33 (17.4)								
Charlson Comorbidity Index *	≥Medium risk	7 (21.2)	1.32	0.52	3.34	0.562				
	Low	27 (17.0)								
CEA (ng/mL)	≥5.0	1 (6.3)	0.29	0.04	2.27	0.210				
	<5.0	33 (18.8)								
LMR ***	<5.08	20 (25.0)	2.56	1.10	4.96	**0.025**	1.63	0.71	3.74	0.254
	≥5.08	14 (12.5)								
CAR ***	≥0.024	13 (25.5)	1.95	0.89	4.27	0.089				
	<0.024	21 (14.9)								
CONUT score	≥Light malnutrition	15 (27.8)	2.41	1.12	5.19	**0.022**	1.81	0.75	4.36	0.184
	Normal	19 (13.8)								
Procedure	TG	12 (30.8)	2.65	1.17	5.99	**0.017**	2.00	0.74	5.39	0.171
	DG, PG, PPG	22 (14.4)								
Approach	Open	16 (25.4)	2.10	1.06	4.47	**0.042**	1.15	0.46	2.84	0.768
	Laparoscopy	18 (14.0)								
Intraoperative blood transfusion	Yes	4 (66.7)	10.40	1.82	59.36	**0.001**	3.90	0.57	26.80	0.166
	No	30 (16.1)								
Pathological stage ***	II, III	7 (16.3)	0.89	0.35	2.18	0.781				
	I	27 (18.1)								

Variables in bold are statistically significant (*p* < 0.05). * Charlson Comorbidity Index was defined by only comorbidities without adjusting age. ** Cut-off values were determined by receiver operating characteristic analysis. *** According to the 15th edition of the Japanese Classification of Gastric Carcinoma. Abbreviations: ASA-PS, American society of anesthesiologists physical status; BMI, body mass index; CAR, C-reactive protein/albumin ratio; CD, Clavien–Dindo classification; CEA, carcinoembryonic antigen; CI, confidence index; CONUT score, controlling nutritional status score; DG, distal gastrectomy; LMR, lymphocyte-to-monocyte ratio; OR, odds ratio; PG, proximal gastrectomy; PPG, pylorus-preserving gastrectomy; TG, total gastrectomy.

**Table 4 cancers-16-00616-t004:** Univariate and multivariate Cox proportional analyses of overall, recurrence-free survival after curative surgery for gastric cancer. (**a**) Overall survival. (**b**) Recurrence-free survival.

**(a)**															
		**Elderly Patient Group (n = 379)**							**Young Patient Group (n = 192)**						
		**Univariate Analysis**			**Multivariate Analysis**				**Univariate Analysis**			**Multivariate Analysis**			
			**5-Year**			**95% CI of OR**				**5-Year**			**95% CI of OR**		
**Variable**	**Categories**	**n (%)**	**Survival**	***p*-Value**	**HR**	**Low**	**High**	***p*-Value**	**n (%)**	**Survival**	***p*-Value**	**HR**	**Low**	**High**	***p*-Value**
ASA-PS	3	37 (9.8)	51.5%	**<0.001**	2.34	1.21	4.55	**0.012**	2 (1.0)	50.0%	**0.014**	2.97	0.23	38.70	0.407
	1, 2	342 (90.2)	85.1%						190 (99.0)	92.4%					
CCI *	≥Medium risk	212 (55.9)	73.6%	**<0.001**	2.90	1.45	5.79	**0.003**	33 (17.2)	84.5%	**0.047**	2.64	0.64	10.93	0.181
	Low	167 (44.1)	92.7%						159 (82.8)	93.5%					
CEA (ng/mL)	≥5.0	49 (13.5)	69.8%	**0.012**	1.67	0.87	3.2	0.121	16 (8.3)	87.1%	0.427				
	<5.0	314 (86.5)	84.6%						16 (91.7)	92.4%					
LMR **	<5.08	197 (52.0)	75.4%	**<0.001**	1.67	0.90	3.07	0.102	80 (41.7)	90.9%	0.530				
	≥5.08	182 (48.0)	89.0%						112 (58.3)	92.7%					
CAR **	≥0.024	164 (43.3)	72.1%	**<0.001**	2.02	1.15	3.56	**0.015**	51 (26.6)	86.2%	0.070				
	<0.024	215 (56.7)	89.3%						141 (73.4)	94.0%					
CONUT score	≥Light malnutrition	166 (43.8)	76.1%	**0.004**	1.08	0.61	1.91	0.787	54 (28.1)	92.6%	0.564				
	Normal	213 (56.2)	86.7%						138 (71.9)	97.1%					
Procedure	TG	92 (24.3)	67.2%	**<0.001**	1.92	1.06	3.50	**0.033**	39 (20.3)	71.1%	**<0.001**	4.28	1.20	15.24	**0.025**
	DG, PG, PPG	287 (75.7)	86.7%						153 (79.7)	97.3%					
Approach	Open	151 (39.8)	71.0%	**<0.001**	1.92	1.00	3.69	**0.049**	63 (32.8)	75.2%	**<0.001**	NA	NA	NA	0.999
	Laparoscopy	228 (60.2)	89.1%						129 (67.2)	100.0%					
Transfusion	Yes	23 (6.1)	60.2%	**0.003**	0.90	0.39	2.09	0.812	6 (3.1)	33.3%	**<0.001**	6.41	1.31	31.44	**0.022**
	No	356 (93.9)	83.4%						129 (96.9)	93.9%					
pStage ***	II, III	98 (25.9)	69.7%	**<0.001**	1.11	0.6	2.04	0.740	43 (22.4)	70.2%	**<0.001**	12.58	2.19	72.18	**0.005**
	I	281 (74.1)	86.0%						149 (77.6)	97.9%					
**(b)**															
		**Elderly Patient Group (n = 379)**							**Young Patient Group (n = 192)**						
		**Univariate Analysis**			**Multivariate Analysis**					**Univariate Analysis**		**Multivariate Analysis**			
			**5-Year**			**95% CI of OR**				**5-Year**			**95% CI of OR**		
**Variable**	**Categories**	**n (%)**	**Survival**	***p*-Value**	**HR**	**Low**	**High**	***p*-Value**	**n (%)**	**Survival**	***p*-Value**	**HR**	**Low**	**High**	***p*-Value**
ASA-PS	3	37 (9.8)	44.7%	**<0.001**	2.51	1.34	4.67	**0.004**	2 (1.0)	50.0%	**0.048**	1.59	0.15	16.52	0.699
	1, 2	342 (90.2)	83.8%						190 (99.0)	90.4%					
CCI *	≥Medium risk	212 (55.9)	71.8%	**<0.001**	2.42	1.29	4.52	**0.006**	33 (17.2)	78.8%	**0.014**	2.13	0.64	7.04	0.217
	Low	167 (44.1)	91.1%						159 (82.8)	92.3%					
CEA (ng/mL)	≥5.0	49 (13.5)	66.3%	**0.005**	1.65	0.90	3.04	0.104	16 (8.3)	87.5%	0.175				
	<5.0	314 (86.5)	83.1%						16 (91.7)	90.2%					
LMR **	<5.08	92 (24.3)	64.1%	**<0.001**	1.72	0.97	3.05	0.064	80 (41.7)	88.5%	0.374				
	≥5.08	287 (75.7)	85.5%						112 (58.3)	91.0%					
CAR **	≥0.024	151 (39.8)	64.1%	**<0.001**	2.51	1.34	4.67	**0.035**	51 (26.6)	84.3%	0.107				
	<0.024	228 (60.2)	85.5%						141 (73.4)	92.0%					
CONUT score	≥Light malnutrition	23 (6.1)	46.7%	**<0.001**	1.33	0.63	2.82	0.451	54 (28.1)	88.7%	0.684				
	Normal	356 (93.9)	82.5%						138 (71.9)	90.4%					
Procedure	TG	98 (25.9)	64.2%	**<0.001**	1.48	0.83	2.62	0.104	39 (20.3)	65.9&	**<0.001**	3.42	1.14	10.26	**0.028**
	DG, PG, PPG	281 (74.1)	85.6%						153 (79.7)	96.1%					
Approach	Open	151 (39.8)	64.1%	**<0.001**	2.51	1.34	4.67	**0.035**	63 (32.8)	65.9%	**<0.001**	7.49	0.84	67.05	0.072
	Laparoscopy	228 (60.2)	85.5%						129 (67.2)	96.5%					
Transfusion	Yes	23 (6.1)	46.7%	**<0.001**	1.33	0.63	2.82	0.451	6 (3.1)	16.7%	**<0.001**	4.82	1.16	20.07	**0.031**
	No	356 (93.9)	82.5%						129 (96.9)	92.4%					
pStage ***	II, III	98 (25.9)	64.2%	**<0.001**	1.48	0.83	2.62	0.104	43 (22.4)	66.7%	**<0.001**	6.58	1.91	22.72	**0.003**
	I	281 (74.1)	85.6%						149 (77.6)	96.6%					

Variables in bold are statistically significant (*p* < 0.05). Categorical variables were described using frequencies and percentages. * Charlson Comorbidity Index was defined by only comorbidities without adjusting age. ** Cut-off values were determined by receiver operating characteristic analysis. *** According to the 15th edition of the Japanese Classification of Gastric Carcinoma. Abbreviations: ASA-PS, American society of anesthesiologists physical status; BMI, body mass index; CAR, C-reactive protein/albumin ratio; CCI, Charlson Comorbidity Index; CEA, carcinoembryonic antigen; CONUT score, controlling nutritional status score; DG, distal gastrectomy; HR, hazard ratio; OR, odds ratio; PG, proximal gastrectomy; pStage, pathological stage; PPG, pylorus-preserving gastrectomy; TG, total gastrectomy.

**Table 5 cancers-16-00616-t005:** Comparison of the variables, according to the CAR of the elderly patient group.

**Elderly (≥65) Patient Group: N = 379**		**Overall Cohort**			**Propensity Score-Matched Pairs**		
**Variables**	**Categories**	**CAR-Low (n = 215)**	**CAR-High (n = 164)**	***p*-Value**	**CAR-Low (n = 143)**	**CAR-High (n = 143)**	***p*-Value**
Sex	male	146 (67.9)	121 (73.8)	0.214	99 (69.2)	102 (71.3)	0.698
	Female	69 (32.1)	43 (26.2)		44 (30.8)	41 (28.7)	
BMI	<18.5	23 (10.7)	22 (13.4)	0.418	18 (12.6)	18 (12.6)	1.000
	≥18.5	192 (89.3)	142 (86.6)	0.420	125 (87.4)	125 (87.4)	
ASA-PS	3	12 (5.6)	25 (15.2)	**0.002**	11 (7.7)	15 (10.5)	0.410
	1, 2	203 (94.4)	139 (84.8)		132 (92.3)	128 (89.5)	
Charlson Comorbidity Index *	≥Medium risk	107 (49.8)	105 (64.0)	**0.006**	89 (62.2)	85 (59.4)	0.628
	Low risk	108 (50.2)	59 (36.0)		54 (37.7)	58 (40.6)	
Diabetes mellitus	present	31 (14.4)	33 (20.1)	0.142	24 (16.8)	26 (18.2)	0.756
	absent	184 (85.6)	131 (79.9)		119 (83.2)	117 (81.8)	
Location of tumor	Upper	53 (24.7)	49 (29.9)	0.256	40 (28.0)	44 (30.8)	0.603
	Middle/Low	162 (75.4)	115 (70.1)		103 (72.0)	99 (69.2)	
Depth of tumor **	T2-4	65 (30.2)	61 (37.2)	0.154	58 (40.6)	46 (32.1)	0.140
	T1a, T1b	150 (69.8)	103 (62.8)		85 (59.4)	97 (67.8)	
Lymph node metastasis **	N1-3	49 (22.8)	51 (31.1)	0.069	42 (29.4)	39 (27.3)	0.155
	N0	166 (77.2)	113 (68.9)		101 (70.6)	104 (72.7)	
Microscopic lymph duct invasion	ly (+)	76 (35.4)	70 (42.7)	0.146	62 (43.4)	58 (40.6)	0.632
	ly (−)	139 (64.7)	94 (57.3)		81 (56.6)	85 (59.4)	
Microvascular invasion	v (+)	52 (24.2)	53 (32.3)	0.080	39 (27.3)	42 (29.4)	0.694
	v (−)	163 (75.8)	111 (67.7)		104 (72.7)	101 (70.6)	
Pathological stage **	II, III	45 (20.9)	53 (32.3)	**0.012**	42 (26.6)	38 (26.6)	0.598
	I	170 (79.1)	111 (67.7)		101 (73.4)	105 (73.4)	
Postoperative complication (≥CD II)	present	48 (22.3)	59 (36.0)	**0.004**	35 (24.5)	51 (35.7)	**0.039**
	absent	167 (77.7)	105 (64.0)		108 (75.5)	92 (64.3)	

Variables in bold are statistically significant (*p* < 0.05). Categorical variables were described using frequencies and percentages. * Charlson Comorbidity Index was defined by only comorbidities without adjusting age. ** According to the 15th edition of the Japanese Classification of Gastric Carcinoma. Propensity score-matching analysis was performed using logistic regression analysis to create a propensity score for the low CAR group and high CAR group with a logistic regression model. The following variables were entered into the propensity model: sex, ASA-PS, CCI, operative procedure, operative approach, pStage. One-to-one matching without replacement was performed with a 0.25 caliper width, and the resulting score-matched pairs were used in subsequent analyses. Abbreviations: ASA-PS, American society of anesthesiologists physical status; BMI, body mass index; CAR, C-reactive protein/albumin ratio; CD, Clavien–Dindo classification; DG, distal gastrectomy; PG, proximal gastrectomy; PPG, pylorus-preserving gastrectomy; TG, total gastrectomy.

## Data Availability

Data are contained within the article.

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
