# Peer review of "Preoperative High C-Reactive Protein to Albumin Ratio Predicts Short- and Long-Term Postoperative Outcomes in Elderly Gastric Cancer Patients"

_cancers, 2024, doi:10.3390/cancers16030616_

Round 1
Reviewer 1 Report
Comments and Suggestions for Authors
I have some major points to discuss :
1/ The interest of a predictive factor is to offer a solution, as changing the definition of surgery, or reject a too much complex surgery, or delay surgery for improving a bad nutritional status, etc …. If no change is possible, as the tumor include a high ratio of “independent small round cell” , my interest is limited.
Another situation is to isolate a factor that affect postoperative outcomes, as Tobacco use, that decrease the rate of wound healing, because of the direct action on the micro vessel density.
Here authors describe a ratio, and consider that it “affect postoperative outcomes”, that is very strange for me. A ratio of two different elements cannot affect something.
Differently, a ratio of two different factors could be a factor associated with a specific risk of postoperative outcome because the ratio summaries a biological information as bad preoperative nutritional status, or specific type of aggressive cancer.
To my opinion, the tittle of the paper is not appropriate and had to be change.
2/ Frailty patients is a recent surgical concept that need to be evaluated in different type of surgical strategy to be able to change or correct the frailty or to change the global strategy. The first and more important group of frailty patient is aged patients.
Regarding that concept it is important to underline any factor that could help to identified specific patient. The first question that had to be evaluated is, can I do a more precise evaluation using specific factors than classical: ASA, OMS, BMI, and recent weight loss for more than 10%, anemia, etc …
The interest of a paper reporting a high number of cases is to compare easy and well-known factors to new one and evaluated if they predicted more precisely the postoperative course.
Authors had to test different prediction models using CARP versus others – for that they need a first group of cases to validate the factor (with only old patients) and a second group of different patients (always old) to confirm the interest in an independent setting.
3/ Inflammation and cancer progression ?
Authors claimed that : “Perioperative inflammatory and nutritional statuses of the host are factors that have been associated with surgical outcomes after some malignancies. The systemic inflammatory response of the host plays an important role in the immune response against acute invasion and tumor progression.” I will be interested to read a paper that study that and confirm the assertion of the authors.
References 13 et 14 are not precise, reported on gastric cancer promotion ? and old.
Some responses are proposed in the discussion chapter, maybe change and include that in the introduction chapter and separate research papers from serious clinical papers on gastric cancer.
A specific sentence in the discussion expose a solution to modify CARP. “Improvements in risk scores may contribute to better outcomes in high-risk patients. Adequate preoperative anti-inflammatory and nutritional treatments can decrease preoperative CAR levels.”
Did authors use anti-inflammatory drugs ? Did they have some reference about that ?
If not they have to suppress that sentence. If yes I will be very interested to read more.
Reviewer 2 Report
Comments and Suggestions for Authors
This manuscript showed that preoperative C-reactive albumin ratio may play a significant role in predicting short- and long-term surgical outcomes in elderly patients, not in young patients with gastric cancer.
The results were acceptable, and the methodology and discussion were well investigated.
In particular, I think it would have been better to consider a large number of cases and analyze them by age.
So we may accept this manuscript without revision.
Comments on the Quality of English LanguageYou need minor change on the quality of English by native speaker.
Author Response
Thank you for your consideration of our manuscript. We appreciate the time spent by you. We are planning to improve the manuscript based on the opinions of you and other reviewers.
Reviewer 3 Report
Comments and Suggestions for Authors
thank you for allowing me to review this retrospective monocentric study which evaluated the impact of albumin/CRP matching in elderly subjects after carcinological gastrectomy.
as the authors point out, the study population was subjects aged 65 and over. i suggest that the authors delete the section on younger subjects, which is irrelevant to the aim of the study and confuses the manuscript. in fact, the propensity score and its results will be more clearly highlighted. similarly, publications using the Dindo-Cavien classification report severe morbidity at 90 days. i suggest that authors analyze risk factors for severe morbidity, which is more relevant at 30 and 90 days.
the authors developed a score based on a value from the CAR report. however, this valuation is not validated or a bootstrap methodology has been developed.
which method was used for the propensity score? inverse probability weighting treatment (IPWT) and propensity score matching (PSM).
with regard to the albumin/CRP ratio (CAR), what was the measurement delta in relation to the intervention?
with regard to the results
as regards the results
Table I attention to errors in the parietal depth of the gastric tumor. as far as I know for stages T2, T3 and T4, the addition of young and old patients is false.
it is recommended in locally advanced gastric tumors to perform perioperative chemotherapy. What treatment did the patients receive? Similarly, it is recommended to renew the undernourished patients in pre-operative. What treatment did these patients receive?
lines 273-280: these are methods and not results.
Given the retrospective nature of the study, what was the rate of missing data?
Round 2
Reviewer 3 Report
Comments and Suggestions for Authors
the authors have responded point by point to the questions and comments that have significantly improved the quality of the manuscript.